# Multistep nucleation of anisotropic molecules

Kazuaki Z. Takahashi [1]✉, Takeshi Aoyagi [1] & Jun-ichi Fukuda [2]

Phase transition of anisotropic materials is ubiquitously observed in physics, biology, materials science, and engineering. Nevertheless, how anisotropy of constituent molecules affects the phase transition dynamics is still poorly understood. Here we investigate numerically the phase transition of a simple model system composed of anisotropic molecules, and report on our discovery of multistep nucleation of nuclei with layered positional ordering (smectic ordering), from a fluid-like nematic phase with orientational order only (no positional order). A trinity of molecular dynamics simulation, machine learning, and molecular cluster analysis yielding free energy landscapes unambiguously demonstrates the dynamics of multistep nucleation process involving characteristic metastable clusters that precede supercritical smectic nuclei and cannot be accounted for by the classical nucleation theory. Our work suggests that molecules of simple shape can exhibit rich and complex nucleation processes, and our numerical approach will provide deeper understanding of phase transitions and resulting structures in anisotropic materials such as biological systems and functional materials.

[1] Research Center for Computational Design of Advanced Functional Materials, National Institute of Advanced Industrial Science and Technology (AIST), Tsukuba, Ibaraki, Japan. [2] Department of Physics, Faculty of Science, Kyushu University, Fukuoka, Fukuoka, Japan. ✉email: kazu.takahashi@aist.go.jp

Phase transition has intrigued researchers not only as an interesting academic problem of condensed matter physics and statistical physics but also from a technological viewpoint[1]. For example, a wide variety of daily products such as food, plastics, and alloys are processed by precise control of the transition from liquid mixture to solid. Phase transition has relevance also to biology because some biological systems such as deoxyribonucleic acid (DNA), ribonucleic acid (RNA), and membranes in our bodies function by repeating many kinds of phase transitions[2–9]. These materials exhibit full of diverseness in structures and functions, and the process of phase transition often determines the performance of products through pattern formation. Hence, in a wide range of research fields including physics, biology, materials science, and engineering, it is an important problem to understand how phase transition occurs and how it can be controlled.

First-order phase transitions occur via the nucleation and growth of the stable new phase from the metastable host phase[1], and typical examples include the transition between the solid phase and liquid phase, and the crystallization of solute molecules from a solution. Nucleation was initially understood by the classical nucleation theory (CNT)[10] in which the Gibbs free energy of a nucleus is assumed to be the sum of the bulk contribution and the interface free energy between the two phases, and evaluated as a function of the size of the nucleus. The distribution of clusters obeys the Boltzmann statistics, and a nucleus starts to grow when its size exceeds a critical value determined by the balance of the bulk and the surface free energies. Although it provided a comprehensible picture of the nucleation and growth processes, CNT was challenged by subsequent experimental studies because it failed to give a quantitative account for the nucleation rate and other pertinent properties of the nucleation processes. Two-step nucleation provides an alternative scenario that could explain the discrepancies between experimental results and the prediction of CNT. Two-step nucleation scenario has been drawing considerable interest, with a successful application to the crystallization of solute molecules, wherein the formation of dense or highly ordered precursors of solute molecules precedes that of crystalline clusters[11–19]. Still, the step-wise pathway of this two-step nucleation can be understood within the context of CNT by considering realistic kinetic factors of clusters[18,19]. An even more complex scenario has been proposed for solute nucleation that does not fall into the category of CNT. In this scenario, the nucleation is typically initiated by the formation of disordered networks or complex frameworks of solute precursors[12,16,19–21] that are thermodynamically stable within the boundary of the mother solution[22,23]. Such clusters are referred to as pre-nucleation clusters (PNCs), although their existence is open to debate[24,25]. The nucleation process believed to involve PNCs evolves in order of the phase separation, condensation, and ripening, and is often referred to as multistep nucleation in a broad sense[26–29].

Here we show by molecular simulations that multistep nucleation takes place in a first-order phase transition of a simple model system composed of a single species of anisotropic particles. Direct real-space observation of the microscopic processes of phase transition, although there have been reports on atomistic[30–32] and colloidal systems[33,34], is still an experimental challenge. Molecular simulations have thus provided an alternative and promising means to elucidate the microscopic mechanisms of phase transition phenomena, in particular nucleation processes. Indeed, the questions of how a crystal nucleus forms in a liquid phase have been addressed for diverse systems[14,15,20,24,25,35–41]. However, there have been few numerical studies that successfully demonstrate the actual non-classical nucleation dynamics together with the energetic stability,

statistics and dynamics of clusters and transient structures involved in the nucleation process. It is because large-scale molecular dynamics simulations involving numerous clusters are required for reliable statistical analyses.

The anisotropy and the resulting additional degrees of freedom of constituent molecules allow a richer possibility of the formation of different ordered phases and thus phase transition behavior, as exemplified in liquid crystals (LCs)[42–44] and also biological systems[6,8,45]. However, this richness renders the investigation of the phase transition behavior even more challenging, although a few studies addressed the formation and kinetic pathway of orientationally ordered clusters from a host isotropic phase[46–48]. Here we focus on pre-transitional clusters with local layered positional order (smectic clusters) formed from a fluid-like nematic phase with orientational order but no positional or layered order. There have been several experimental studies that corroborated the formation of pre-transitional smectic clusters (also known as cybotactic clusters) by X-ray diffraction[49,50]. However, few studies address the question of how such pre-transitional fluctuations and smectic nuclei are formed. In molecular simulation, the difficulty lies in identifying local smectic order from orientationally ordered nematic phase, which is overcome by supervised machine learning (ML) that finds appropriate order parameters for this purpose[51]. In this work, we show the power of the trinity of molecular dynamics (MD) simulations, ML and molecular cluster analysis by investigating the elusive dynamics of the formation of smectic clusters from a nematic phase, and spotting the multistep nature of its kinetic pathways.

## Results

We performed MD simulations of 1 million particles of Soft-Core Gay-Berne (SCGB) model[52,53]. Well equilibrated nematic systems were quenched to temperature $T = 1.80$, below the smectic transition temperature $T_{N-Sm} = 2.25$[53] (See Methods for the definition of $T$). The time series of quenched coordinates was then analyzed using the above-mentioned ML scheme that precisely determines whether a certain molecule belongs to a nematic-like or a smectic-like local structure (for details, refer to Methods, and Supplementary Fig. 1). A molecular cluster analysis was applied to the time series of extracted smectic-like local structures.

**Behavior of the whole system**. To observe how the smectic ordering evolves in the whole system during phase transition, the time evolution of the number of "smectic molecules" was traced, as shown in Fig. 1a. The number of smectic molecules belonging to the largest smectic cluster is also plotted. Smectic nucleation must precede the drastic increase of molecules belonging to the largest smectic cluster around $t \simeq 1.0\tau$ ($\tau$ is the time unit associated with a single particle, described in Methods). The percolation of the smectic phase progresses at $1.0\tau \lesssim t \lesssim 1.5\tau$, until almost all the smectic molecules belong to the largest cluster at $t \gtrsim 1.5\tau$.

Previous X-ray scattering experiments[49,50] suggest the formation of pre-transitional fluctuations, known as cybotactic clusters, at a relatively early period of the transition to the smectic phase. In Fig. 1b we show the time evolution of numerical X-ray scattering intensity that can be directly compared with experiments (for calculation details and scattering intensity profiles, refer to Supplementary Fig. 2). The scattering intensity is plotted also as a function of the number of smectic molecules in the system. Figure 1b clearly indicates that the scattering intensity depends almost linearly on both the time ($\lesssim 1.0\tau$), and the number of smectic molecules in the system ($\lesssim 1.7 \times 10^5$),

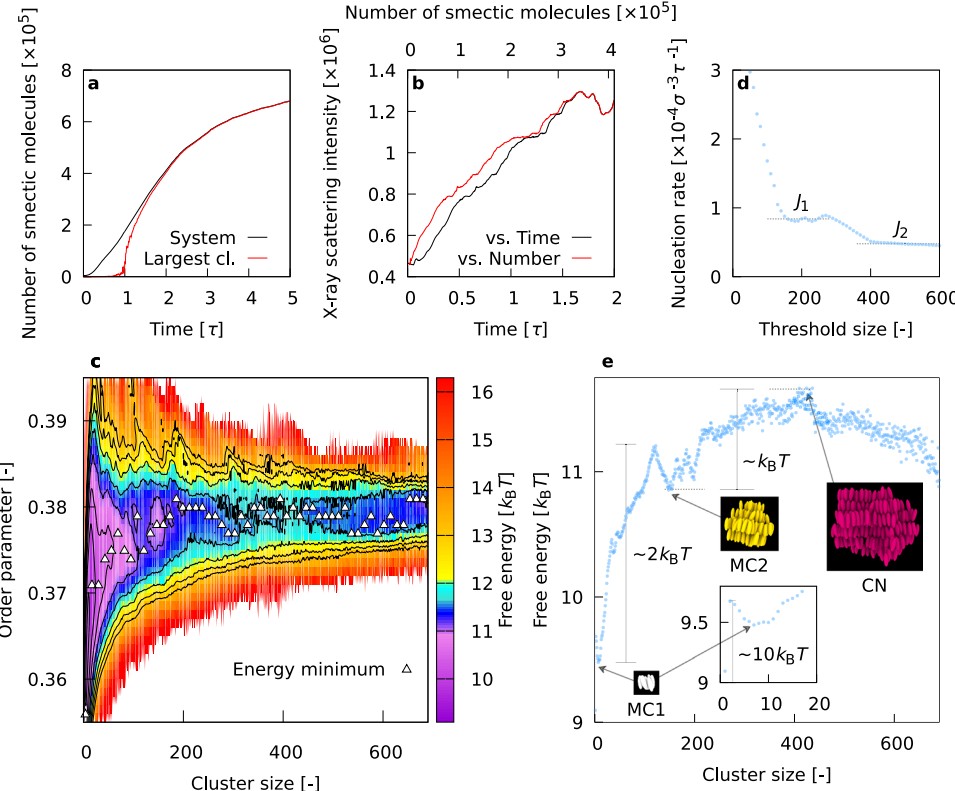

**Fig. 1 Behavior of the whole system and free energy landscape. a** Time evolution of the number of smectic molecules in the system. The number of smectic molecules belonging to the largest smectic cluster is also plotted. **b** Time evolution of numerical X-ray scattering intensity. The intensity as a function of the number of smectic molecules in the system is also plotted. **c** Contour map of free energy landscape as a function of smectic cluster size and order parameter value. The black lines indicate contours with interval of 0.5 $k_B T$. **d** Estimated value of the nucleation rate as a function of the threshold size. $J_1$ and $J_2$ are plateau values. **e** Free energy minima as a function of the cluster size.

suggestive of a strong correlation between the X-ray scattering intensity and the number of smectic molecules determined by our ML scheme. As noted above, smectic nucleation dominates the kinetic process of phase transition at $t \lesssim 1.0\tau$. Hence, the pre-transitional local structures or cybotactic clusters involved in the smectic nucleation process are indeed responsible for the growth of the X-ray scattering intensity observed in experiments.

**Free energy landscape**. To obtain more detailed information on the static and dynamic properties of pre-transitional local structures, we constructed the free energy landscape as a function of the cluster size $N$ and the order parameter quantifying the degree of liquid crystalline order. The order parameter $Q$ was provided by our ML scheme[51,54], and the free energy landscape was calculated by the transition probability approach based on the previous work of Mochizuki and co-workers[55] (for calculation details, refer to Methods). The latter is presented in Fig. 1c (the reference (zero) of the free energy is taken to be that of the nematic phase), and the saddle point corresponds to the critical nucleus that is made up of approximately 420 smectic molecules.

Furthermore, a pocket region at $N \simeq 150$ was discovered, indicating the existence of metastable clusters. This estimate of the size of the critical nuclei as $N \simeq 420$, and the presence of metastable clusters at $N \simeq 150$ agree well with the result of a conventional threshold method[56] shown in Fig. 1d that focuses only on the size of the critical nuclei (for calculation details, refer to Supplementary Fig. 3). In Fig. 1c, each white rectangle highlights the free energy minimum for a given cluster size, and the sequence of white rectangles specifies the major pathway of the nucleation

process. The pathway starts with a drastic increase of the order parameter of small clusters and reaches the saddle point via the pocket region. Initial ordering of small clusters before passing the saddle point of the free energy landscape is characteristic of two-step nucleation processes[14,18,57]. However, the presence of a pocket region can make the nucleation process unique and more complex than the conventional two-step scenario.

To understand more simply the major pathway of nucleation, the free energy minima highlighted in Fig. 1c are replotted in Fig. 1e as a function of the cluster size. From Fig. 1e, two metastable clusters can be identified around $N = 7$ and $N = 150$, labeled MC1 and MC2, respectively. The free energy barrier from MC1 to MC2 is $\sim 2k_B T$, and that from MC2 to critical nuclei (CN) is $\sim k_B T$. The total free energy barrier of the major pathway from the nematic phase (reference state) is $\sim 11.7 k_B T$, consistent with the fact that the nematic-smectic phase transition is weakly first order[58,59]. The universality of the formation of two metastable clusters corresponding to MC1 and MC2 irrespective of the model and the initial structures is demonstrated by the simulations using the original Gay-Berne (GB) model[60], as shown in Supplementary Fig. 4.

We carried out further systematic calculations to elucidate how the variation of the degree of supercooling $\Delta T \equiv T_{\text{N-Sm}} - T$ influences the free energy landscape and the resulting major pathway and metastable clusters. The same qualitative features described above are observed regardless of $\Delta T$ (see Supplementary Fig. 5). In Fig. 2a, b we show the $\Delta T$ dependence of the CN size and the height of the energy barrier from nematic to CN, respectively. Figure 2 clearly demonstrates the non-classical nature of the nematic-smectic phase transition; the CN size is

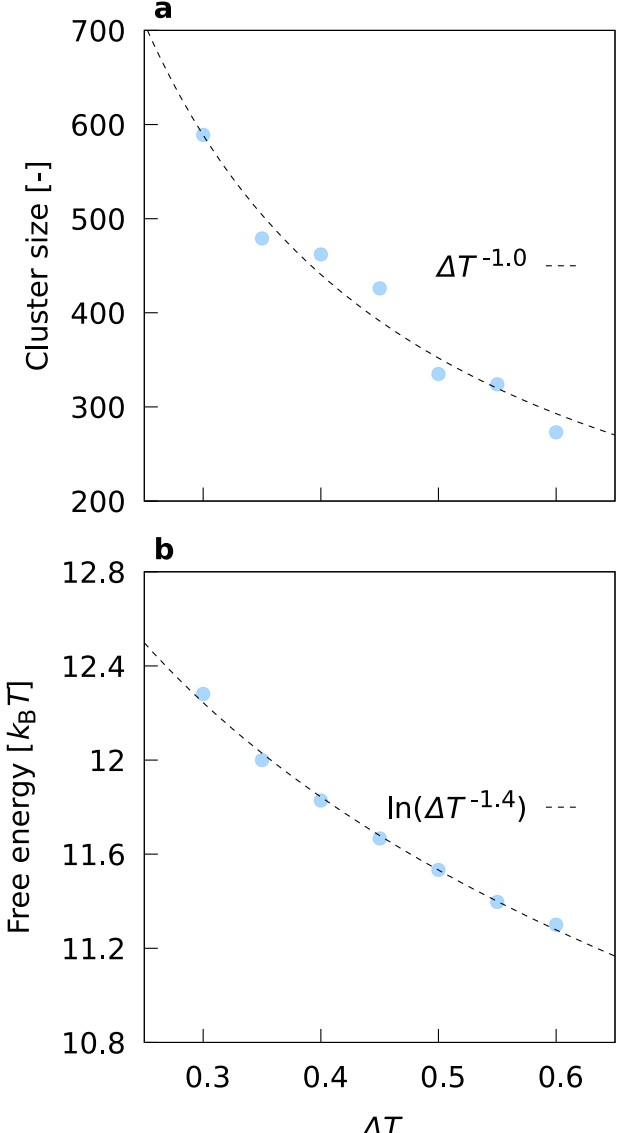

**Fig. 2 Scaling behavior of smectic nucleation.** $\Delta T$ dependence of **a** the CN size and **b** the height of the energy barrier from nematic to CN.

MC2) and CN (More details of the identification criteria are presented in Methods, and Supplementary Fig. 7), their real-space geometrical information can be deduced. Figure 3 shows the averaged density profiles of MC1, MC2, and CN. The *abc* orthogonal coordinate system is such that its origin is at the center of the molecule that is closest to the center of mass of the cluster, and the *a*-axis is parallel to the average orientation of the constituent molecules. The void at the origin indicates that the molecule at the cluster center excludes the other molecules sterically. MC1 shown in Fig. 3a is composed of a simple hexagonal column and has a small tail in the direction perpendicular to the *a*-axis. The shape of MC2 shown in Fig. 3b is oblate, although it may depend on the interaction parameters as suggested in a previous study on nematic droplets[48]. Figure 3b clearly indicates that at least three distinct smectic layers are involved, with additional 1 or 2 layers being recognized as the tail along the *a* direction. The shape of CN shown in Fig. 3c, involving 5 distinct smectic layers and additional 1 or 2 layers as the tail, is highly similar to that of MC2. We emphasize that the real-space profile of pre-transitional fluctuations towards the smectic phase, also known as cybotactic clusters, has been presented only schematically[61]. Our work, with the aid of the trinity of MD simulation, ML, and molecular cluster analysis, provides the first real-space density profiles of metastable clusters and critical nuclei in a clear manner.

**Dynamics of metastable clusters, critical nuclei and supercritical nuclei in the transition**. For further understanding of the smectic nucleation process, it is important not only to construct the free energy landscape as discussed above, but also to observe the actual dynamics of smectic clusters associated with the major pathway of the free energy landscape. We have already presented how MC1, MC2 and CN are identified, and we also define a supercritical nucleus (ScN) as a cluster larger than CN (namely, $N > 443$, as the size of CN is $410 \leq N \leq 442$. See Supplementary Fig. 7c). To monitor the history of the clusters, we further introduce the labelings tMC1, tMC2, and tCN; We label as tMC1 a cluster or a molecule which was formerly MC1 but does not belong to MC1, MC2, CN, nematic phase or ScN at the time of the identification. The labelings tMC2 and tCN are defined similarly. The size distribution of the clusters labeled as MC1, MC2, CN, tMC1, tMC2, or tCN shown in Supplementary Fig. 8, whose vertical axis is the logarithm of the existence probability, agrees well with minus the free energy landscape in Fig. 1e. Recalling that Fig. 1e is the major pathway of the nucleation process, and that minus the logarithm of the size distribution represents the free energy as a function of the cluster size, our identification of characteristic molecules or clusters (MC1, MC2, CN, tMC1, tMC2 or tCN) successfully samples the major pathway of the nucleation process.

Now we focus on how ScNs are formed; more specifically, whether a given ScN emerges through the major pathway of the nucleation process. We refer to ScNs formed through the major pathway as "induced ScN" (IScN), and those through other minor pathways as "non-induced ScN" (NScN). The identification of IScN and NScN is as follows: We count the number of characteristic molecules included in newly born ScNs during nucleation process. We denote by $M_t$ the sum of the numbers of characteristic molecules included in an ScN. From the bipolar form of the generation probability of ScNs as a function of $M_t$ (Supplementary Fig. 9), we identify an ScN as IScN if $M_t \geq 200$, and as NScN otherwise. Note that in the following analyses the characteristic molecules in newly born IScNs or NScNs are immediately relabeled as IScN or NScN. More specific rules for the classification of molecules are shown in Supplementary Fig. 10.

proportional to $\Delta T^{-1.0}$, and the barrier from nematic to CN is proportional to $\ln(\Delta T^{-1.4})$ (In CNT, they should behave as $\propto \Delta T^{-3}$ and $\propto \Delta T^{-2}$, respectively). The MC1 and MC2 sizes are almost independent of $\Delta T$ (respectively 7 and $168 \pm 20$, see Supplementary Fig. 6a). The barrier from MC1 to MC2 decreases weakly with the increase of $\Delta T$ and falls within the range $[1.60k_BT, 2.22k_BT]$ (see Supplementary Fig. 6b). The barriers from nematic to MC1 and from MC2 to CN are almost constant $((9.68 \pm 0.04)\,k_BT$ and $(0.77 \pm 0.08)\,k_BT$, respectively). The depth of the local minimum of MC1 and MC2 is only weakly dependent of $\Delta T$ and increases with the increase of $\Delta T$ (see Supplementary Fig. 6c). The high barrier $\simeq 9.68\,k_BT$ clearly indicates that MC1 is metastable. From the low energy barrier from MC1 ($\sim 2.0k_BT$), and the size insensitivity to $\Delta T$, MC2 might be regarded as "metastable PNC" formed in a mixture of host nematic phase and MC1 clusters. In view of the insensitivity of the features of metastable clusters to $\Delta T$, in the following sections $T$ is set to 1.80 except where specifically noted.

**Real-space density profiles of metastable clusters and critical nuclei**. After the identification of metastable clusters (MC1 and

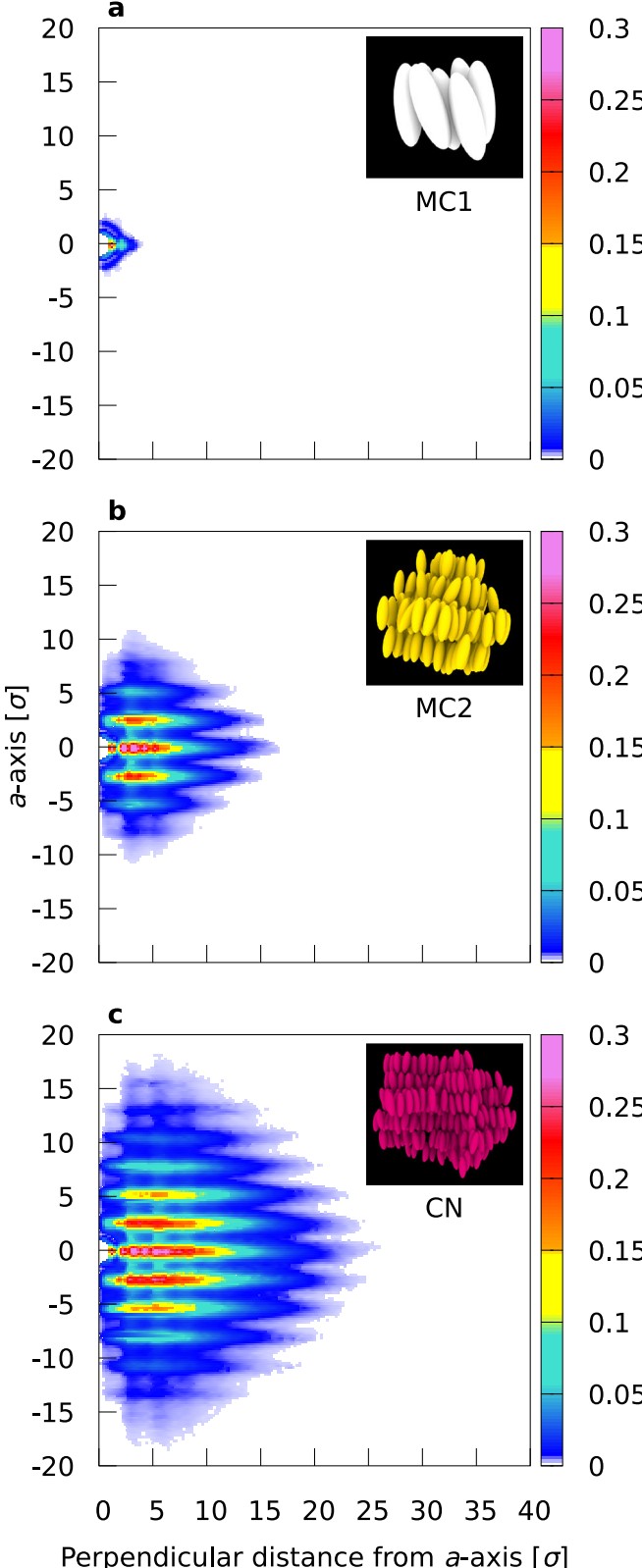

**Fig. 3 Real-space profiles of metastable clusters and critical nuclei.**
Density profiles of **a** MC1, **b** MC2, and **c** CN. The density is circularly
averaged about the $a$-axis that is parallel to the average orientation
of the constituent molecules, and the horizontal axis is the distance
to the $a$-axis.

IScNs become the largest component of the system along with
the progress of the nucleation process (Supplementary Fig. 11), and
therefore monitoring the time evolution of the number of IScNs is
highly important for the understanding of the smectic nucleation
process. Figure 4a clearly demonstrates that the nucleation of IScNs
involves three processes: The first process is at $0.50\tau < t < 0.60\tau$
with the nucleation rate of $J_{1st} = 2.26 \times 10^{-5}\sigma^{-3}\tau^{-1}$, followed by
the 1st plateau. The second process is at $0.64\tau < t < 0.69\tau$ with the
nucleation rate of $J_{2nd} = 6.28 \times 10^{-5}\sigma^{-3}\tau^{-1}$, followed by the 2nd
plateau. The latter is conceivable as the lag time to the third process
at $0.75\tau < t < 0.80\tau$ with the nucleation rate of $J_{3rd} = 7.61 \times$
$10^{-5}\sigma^{-3}\tau^{-1}$, again followed by the 3rd plateau. Note that the
nucleation rate increases with the evolution of the nucleation
processes. In the following, we refer to these three processes as the
"first (second or third) nucleation process".

Let us consider the origin of three-step nucleation. The first
nucleation rate is clearly smaller than that for the other processes,
and is close to the nucleation rate for the minor pathway (see
Supplementary Fig. 12). Hence the 1st process is a nucleation
before forming the efficient pathway. The time evolution of the
number of tCNs and tMC2s shown in Fig. 4b explains the second
and third processes. The number of tCNs has two peaks; it
steadily increases until the end of the 2nd nucleation process and
then starts to decrease, followed by the second increase during the
2nd plateau. Its second peak is at the end of the 3rd nucleation
process. Hence these two peaks obviously mark the 2nd and 3rd
nucleation processes of IScNs. Note that tCNs are the largest
component of characteristic molecules included in IScNs, and
clearly dominate the formation of IScNs (Supplementary Fig. 13).
The number of tMC2s exhibits one clear peak at the beginning of
the 2nd nucleation process followed by the decrease. The recovery
of the number of tCNs during the 2nd plateau is clearly attributed
to the consumption of tMC2s to form CNs and thus tCNs.
Therefore, the lag time between 2nd and 3rd nucleation arises
from the competition between the consumption of tCNs due to
fast nucleation and the supply of tCNs from tMC2s.

The importance of tMC2 in the nucleation process is
demonstrated also by monitoring the time evolution of the
formation of characteristic clusters in real space. Figure 4c, d
shows the snapshots of tMC2s (yellow), tCNs (red) and IScNs
(blue) at $t = 0.50\tau$ and $0.80\tau$, respectively. The positional overlap
between tMC2s at $t = 0.50\tau$ and IScNs at $t = 0.80\tau$ clearly
indicates that the former induces the nucleation of the latter. In
fact, the overlap rate is consistently high, and steadily increases at
$t > 0.50\tau$ (Supplementary Fig. 14). The low mobility of tMC2 is
clearly seen in Supplementary Fig. 15 and Supplementary Movie 1,
which reflects the trapping effect owing to the local minimum of
the free energy. Hence low-mobility regions of smectic molecules
constituting tMC2 form the backbone for the nucleation of IScNs,
which highly resembles the formation of crystal ice from the low-
mobility regions of supercooled liquid water[40].

The second and third nucleation processes on the major
pathway have been shown to proceed in the order of MC2, CN,
and IScN, and therefore cannot be regarded as one- or two-step
processes. MC2 and tMC2 distinguish the multistep nucleation
process of the nematic-smectic transition from conventional ones,
and as mentioned above, play an important role in that they form
the backbone for efficient pathway, and that tMC2 clusters act
as the precursors for the nucleation of IScNs. We therefore
conclude that MC2 and tMC2 are unambiguously identified as
the pre-transitional fluctuations that dominate the dynamics of
nematic-smectic transition by inducing CN and tCN, and
thus IScN.

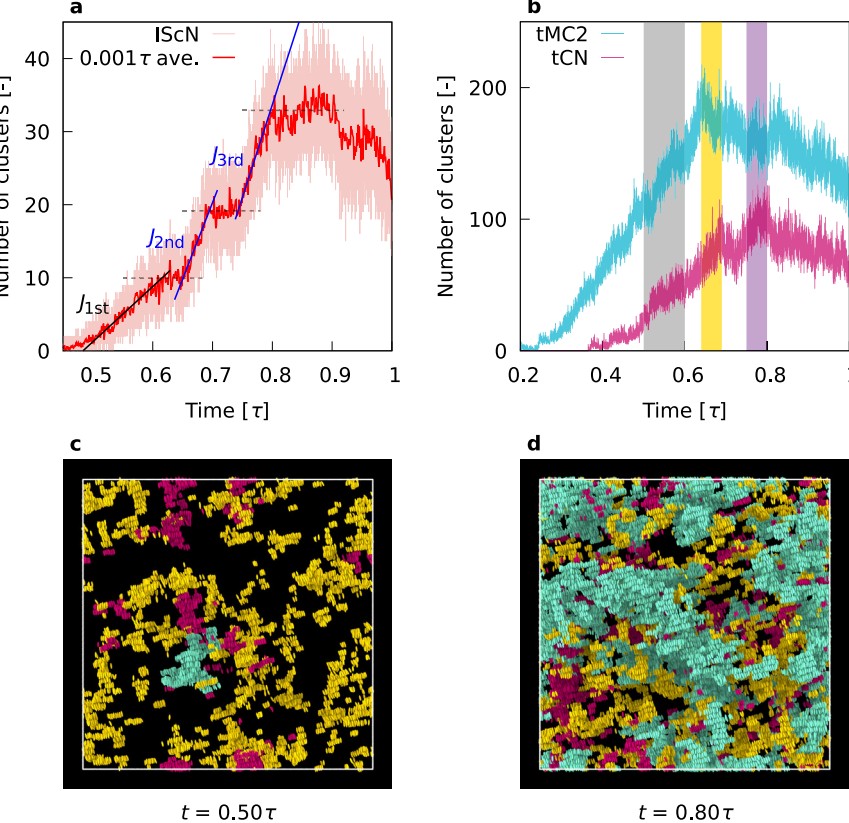

**Fig. 4 Multistep nucleation. a** Time evolution of the number of IScNs. The averaged number of clusters per $0.001\tau$ is also plotted as the solid red line. The dashed lines indicate the plateaus. **b** Time evolution of the number of tCNs and tMC2s. Gray, yellow and violet areas indicate the time range of the first, second and third nucleation, respectively. **c, d** Snapshots of tMC2s (yellow), tCNs (red) and IScNs (blue) at $t = 0.50\tau$ and $0.80\tau$, respectively.

## Discussion

We investigated numerically how anisotropic molecules attain their positional order, or more specifically, how smectic nuclei are formed in the course of the nematic-smectic phase transition. Our state-of-the-art numerical techniques, namely, the trinity of molecular dynamics simulation, machine learning, and molecular cluster analysis, discovered three-step nucleation involving two different types of metastable clusters with properties distinct from those predicted by the CNT. The origin of the three-step nucleation was explained by clarifying the free energy landscape and the major pathway of the nucleation process, and also by tracking the dynamics of metastable clusters and CNs. Our analysis also revealed the positional overlap between metastable clusters and ensuing IScNs, and the observation of metastable clusters by our numerical techniques might enable the prediction of where pattern formation would be initiated, without monitoring the whole nucleation processes. Our demonstration of multistep nucleation in a simple one-component system of model anisotropic molecules has a profound meaning because nucleation processes of simple systems, not restricted to anisotropic ones, can be far richer and more complex than previous studies have shown[62], and we hope our study will promote further studies towards deeper understanding of the complex nature of nucleation phenomena.

Our approach is particularly promising for the investigation of phase transition behavior of biological LCs that widely exist in nature as a rich set of soft materials with anisotropic structures. By tracking metastable clusters, sophisticated functions related to phase transition might be revealed for in vivo or in vitro structures of biological building blocks[2,6–8,45,63]. Our approach also has a great potential for understanding not just phase transition

of LCs but a much broader range of phase transition phenomena in solutions and polymers. More broadly, suitable molecular design that can optimize the size and shape of metastable clusters may enable the control of nucleation rates and pattern formation in a wide variety of anisotropic materials. This is significant not only for the control of basic properties, but also for the design of emergent properties of advanced materials such as self-healing[64]. Further investigations of phase transition phenomena with detailed analysis of energetic stability, structure and dynamics of clusters will open the door to new technology for designing highly advanced materials.

## Methods

**Molecular dynamics simulations.** The MD simulations of GB and SCGB particle systems were performed using an open source program called the Large-scale Atomic/Molecular Massively Parallel Simulator (LAMMPS)[65].

For the intermolecular interactions of ellipsoidal GB particles, the following pairwise interaction potential, $U_{GB}$, was calculated for each pair of particles:

$$U_{GB} = 4\varepsilon_a \left[ \left( \frac{\sigma_s}{r - \sigma_a + \sigma_s} \right)^{12} - \left( \frac{\sigma_s}{r - \sigma_a + \sigma_s} \right)^6 \right], \quad (1)$$

where $\varepsilon_a$ denotes the anisotropic energy for an ellipsoidal pair, $r$ the distance between the centers of mass for a pair of particles, $\sigma_a$ the anisotropic length for the ellipsoidal pair, and $\sigma_s$ the length for the side-by-side configuration of the ellipsoids. By introducing parameter $\kappa = \sigma_e/\sigma_s$, in which $\sigma_e$ denotes the length for the end-to-end configuration of the ellipsoids, the anisotropic energy $\varepsilon_a$ is then written

$$\varepsilon_a = \varepsilon(\varepsilon_a')^\mu(\varepsilon_a'')^\nu, \quad (2)$$

where $\varepsilon$ denotes the characteristic well depth of the interaction potential, $\varepsilon_a'$ and $\varepsilon_a''$ denote the contributions corresponding to the well depth and configuration anisotropies, and $\mu$ and $\nu$ are multipliers for determining these two contributions for the pair potential. Factor $\varepsilon_a'$ is characterized by introducing a parameter $\kappa' = \varepsilon_s'/\varepsilon_e'$, where $\varepsilon_e'$ and $\varepsilon_s'$ denote energy contributions from the end-to-end and

side-by-side configurations of ellipsoids, respectively. Factor $\varepsilon_a''$ is characterized by $\kappa$. Therefore, the detailed shape of $U_{GB}$ is determined from the values of four parameters $\kappa$, $\kappa'$, $\mu$, and $\nu$. For the characteristic length, energy, and mass of the GB systems, $\sigma = \sigma_s$, $\varepsilon = k_B T$, and $m$ are each set to 1; here $m$ is the mass of one GB particle.

For the intermolecular interactions of SCGB particles, the core repulsion of the original GB potential is replaced by a weaker linear repulsion. The soft-core potential energy $U_{SC}$ is written

$$U_{SC} = a(r - \sigma_a), \tag{3}$$

where $a$ is the potential slope for soft repulsive energy barrier. The $U_{GB}$ and $U_{SC}$ are merged using a sigmoidal logistic function $f$ so that the resulting potential reads

$$U_{SCGB} = (1 - f)U_{GB} + fU_{SC}, \tag{4}$$

$$f = 1/\{1 + \exp[b(\sigma_a - r)]\}, \tag{5}$$

where $b$ is the steepness of sigmoidal logistic seaming function.

To be able to compare the results of GB and SCGB systems directly, we used exactly the same parameter settings of the GB potential for the two models. Specifically, $\kappa = 3$, $\kappa' = 5$, $\mu = 1$, and $\nu = 3$. Note that the above parameter set has been traditionally used because the physical properties of nematic and smectic liquid crystal phases are well displayed[53,60]. For SCGB, the terms $a$ and $b$ were set to $-70\varepsilon\sigma^{-1}$ and $-100\sigma^{-1}$, respectively, on the basis of previous reports[52,53]. Using the above parameter settings, the nematic-smectic transition is guaranteed to be observed for both GB and SCGB systems by quenching from temperature $T = 2.4$ to 1.8 at the density of $0.3\sigma^{-3}$ [53]. Therefore, an ensemble was used having a constant number of particles at density $0.3\sigma^{-3}$ contained within a cubic box of constant volume and temperature and with full periodic boundary conditions imposed. The initial configurations for the nematic-smectic phase transition trajectory were prepared in a careful manner, being cooled gradually from the isotropic phase at $T = 6.0$ to the nematic phase at $T = 2.4$. The configurations were then quenched to temperature below $T_{N-Sm}$. To observe the fast nucleation during the weak first-order phase transition, the velocity Verlet integrator with fine timesteps $\Delta t = 6.0 \times 10^{-5}\tau$ and $2.0 \times 10^{-5}\tau$ was used for GB and SCGB, respectively, where $\tau = \sigma(m/\varepsilon)^{1/2}$ is a time unit. The temperature was controlled using a Nosé–Hoover chain thermostat[55]. The velocity-scaling method was also confirmed by yielding consistent results. To compute precisely the intermolecular interactions during phase transition, the GB and SCGB potentials were truncated at $8.0\sigma$.

For precise computations of the free energy landscape (for details, refer to Calculation of free energy landscape), a large number of smectic clusters must be sampled. Therefore, additional MD simulations of the GB and SCGB systems were performed using the smaller timesteps of $\Delta t = 1.5 \times 10^{-5}\tau$ and $0.5 \times 10^{-5}\tau$, respectively, while maintaining other simulation settings described above fixed.

To explore the $\Delta T$ dependence of the free energy landscape of SCGB systems, $T$ was varied from 1.65 to 1.95 (i.e., $\Delta T$ from 0.30 to 0.60).

**Machine learning.** The time series of quenched coordinates was analyzed using the Machine Learning-aided Local Structure Analyzer (ML-LSA)[51,54]. Supplementary Fig. 1 shows the ML flow of ML-LSA specialized for this work. To consider the classification capability of the enormous variety of local order parameters, the flow was designed using simple ML techniques. First, well-defined structure motifs of the nematic and smectic phases were prepared from MD simulations of 1701 (SC) GB particles. Second, the local structure of particle $i$, $L_i$, was defined from the set of particles around $i$. Up to 24 neighbors were considered for $L_i$. The 340,200 $L_i$s were sampled for both phase structures. Third, over 1 million local order parameters derived from the eleven different functions[51] were computed for a total of 680,400 local coordinates as structure descriptors. Each local coordinate was also tagged with a well-defined structure name (nematic or smectic) as the response variable. Fourth, the structure descriptors and structure names were merged with a descriptor array $D$ and a structure name vector $n$, respectively. Fifth, the operator vector $w$ satisfying the relation $Dw = n$ was estimated through ML. The term $w$ was estimated using the random forest method[66] implemented on Scikit-learn[67]. A decision tree of depth 10 was used for the random forest. The $w$ was checked via a $k$-fold cross validation implemented on Scikit-learn for checking overlearning, where $k$ denotes the number of times cross validation is performed. We set $k = 5$ taking into account the quality and quantity of our data in this work. Specifically, 1/5 of 680,400 local coordinates were used for each of the five cross validations. The classification accuracy can be rigorously estimated in terms of the correct tagging rate $C$, expressed as

$$C = \frac{Z_{correct}}{Z_{total}}, \tag{6}$$

where $Z_{correct}$ denotes the number of correct tags derived from $Dw$, and $Z_{total}$ the total number of tags. Note that checking whether each tag is correct is a trivial task because all of the correct tags $n$ were in hand. Actually, the ML scheme was used for developing the best single local order parameter required for the high-performance reaction coordinate describing the nematic-smectic phase transition in this work. The best is a modified bond-orientational order parameter

considering the first to twelfth neighbors, $Q_{l=2}(i)$, defined as follows:

$$Q_{l=2}(i) = \frac{1}{13} \sum_{j \in \tilde{N}_b(i)} \bar{q}_{l=2}(j), \tag{7}$$

$$\bar{q}_{l=2}(i) = \sqrt{\frac{4\pi}{5} \sum_{m=-2}^{2} |\bar{q}_{l=2,m}(i)|^2}, \tag{8}$$

$$\bar{q}_{l=2,m}(i) = \frac{1}{13} \sum_{j \in \tilde{N}_b(i)} q_{l=2,m}(j), \tag{9}$$

$$q_{l=2,m}(i) = \frac{1}{12} \sum_{j \in N_b(i)} Y_{l=2,m}(\mathbf{r}_{ij}), \tag{10}$$

where $l$ is an arbitrary positive integer denoting the degree of the harmonic function, $m$ an integer that runs from $-l$ to $+l$, $\tilde{N}_b(i)$ an array that includes the identification number of particle $i$ itself and those of all its neighboring particles, $N_b(i)$ an array of identification numbers for all neighboring particles around particle $i$, $Y_{l,m}$ the spherical harmonic function, and $\mathbf{r}_{ij}$ the vector from particle $i$ to $j$. The local order parameter $Q_{l=2}(i)$ shows the best classification accuracy $C > 0.996$, regardless of the model difference between GB and SCGB. The performance of $Q_{l=2}(i)$ was unrivaled at least in comparison with conventional local order parameters and their combinations (see Supplementary Fig. 16). Finally, the ML training results of $w$ were applied to classify the nematic- and smectic-like local structures of the quenched systems. Specifically, the structure name vector of the quenched systems, $n_q$, was determined using a two-step procedure; (i) creating the descriptor array of the quenched systems, $D_q$, and (ii) computing $D_q w$ as an approximation of $n_q$. Note that $D_q$ is assumed to be a function of time, and $w$ a time-independent constant. Therefore, $n_q$ can also be considered as a function of time, making it possible to observe the time evolution of nematic- and smectic-like local structures during a phase transition. Note also that a name and order parameter value of the local structure are assigned to each particle, making it possible to apply in a particle-based structure analysis. The ML scheme has already succeeded in classifying the local structures of the LC polymers, having many complicated interfaces between local structures[51]. Therefore, the scheme is sufficiently reliable for our present purpose as well. For further details of ML-LSA and the scheme using it, we refer to our previous work[51,54,68,69].

**Molecular cluster analysis.** To observe smectic nucleation in a nematic-smectic phase transition, a molecular cluster analysis was applied in a time-series analysis of smectic-like local structures extracted from the ML scheme. In the cluster analysis, a cluster was defined as a group of mutually connected molecules within the region containing the first to twelfth neighbor molecules defined in the ML scheme. For the order parameter of the smectic clusters, the averaged local order parameter $Q$ was calculated from $Q_{l=2}(i)$'s belonging to the same cluster. The states of clusters were defined as a function of $Q$ and cluster size $N$.

**Calculation of free energy landscape.** The free energy landscape as a function of $N$ and $Q$ was calculated using the transition probability approach[48,55], which considers a network of states of clusters connected by transition paths with certain transition probabilities. The state of a cluster is defined by $s = \{N, Q\}$. Let $p(t_1, s_1)$ denote the probability of a cluster being in a state $s_1$ at time $t_1$. Then

$$p(t_2, s_2) = \sum_{s_1} p(s_2|s_1)p(t_1, s_1), \tag{11}$$

where $p(s_2|s_1)$ is the transition probability from state $s_1$ at time $t_1$ to $s_2$ at time $t_2$ ($> t_1$), which can be obtained from cluster statistics. Hence the stationary distribution $p(s)$ should satisfy

$$p(s_2) = \sum_{s_1} p(s_2|s_1)p(s_1). \tag{12}$$

When $p(s_2|s_1)$ is given, $p(s)$ can be determined from the iteration of Eq. (11). The free energy landscape $\Delta G(N, Q)$ was obtained from $\Delta G(N, Q) = -k_B T \ln[p(N, Q)]$, where $p(N, Q) = p(s)$. The transition probability approach requires constant transition probabilities among cluster states, except those considerably larger than the critical nuclei[55]. For the quality of the statistics, the time interval $\Delta t_{samp}$ for the sampling should be as large as possible. The optimum $\Delta t_{samp}$ was determined by comparing the results with different $\Delta t_{samp}$.

**Identification of MC1, MC2, and CN.** Here, we describe how metastable clusters (MC1 and MC2) and critical nuclei (CN) were identified using the free energy landscape in terms of the cluster size and the order parameter. The state of a specific cluster was identified by the limited range in free energy, cluster size, and order parameter values. Here we determined the range in free energy and cluster size using the free energy minimum curve illustrated in Fig. 1e, and therefore the range of the order parameter was automatically determined. For MC1 or MC2, the corresponding local minimum in Fig. 1e was selected as a reference state, and the half height of the nearest and steepest wall in Fig. 1e defines the threshold value for energy. States below the threshold were regarded as being from the same metastable cluster. For CN, all points near the largest energy value with no varying trend

with respect to cluster size were regarded as the CNs. Supplementary Fig. 7 shows the free energy contour map of all the states of MC1, MC2, and CN, selected by the above identification criteria.

**Reporting summary**. Further information on research design is available in the Nature Research Reporting Summary linked to this article.

## Data availability

The data that support the findings of this study are available from the corresponding author upon reasonable request, based on the publication protocol of the research data as permitted by a project (JPNP16010) commissioned by the New Energy and Industrial Technology Development Organization (NEDO).

## Code availability

ML-LSA and other codes for analysis are available from the corresponding author upon reasonable request, based on the publication protocol of the developed codes as permitted by JPNP16010 commissioned by NEDO.

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

## Acknowledgements

The authors are grateful to Dr. Hideo Doi of AIST for supporting the use of ML-LSA. We also thank Dr. Ryuji Sakamaki and Mr. Yuki Furukawa of X-Ability Co., Ltd. for computational optimization of GB and SCGB models in molecular dynamics simulations. This work is based on results obtained from JPNP16010 commissioned by NEDO. J.F. is in part supported by JSPS KAKENHI (Grant Numbers JP17H02947 and JP21H01049).

## Author contributions

K.Z.T. and J.F. designed the research; K.Z.T. carried out molecular dynamics simulations, machine learning, and molecular cluster analysis. K.Z.T. and J.F. drafted the paper. T.A. contributed to the interpretation of the part of results and to the writing.

## Competing interests

The authors declare no competing interests.
