## [Peer Review File · Nature Communications]

Multistep nucleation of anisotropic moleculesREVIEWER COMMENTS

Reviewer #1 (Remarks to the Author):

The manuscript of Takahashi et al is very well written, dealing with the multistep nucleation of anisotropic molecules, which was explored computationally. Namely, the authors used a combination of molecular dynamics simulation, machine learning, and molecular cluster analysis to obtain very interesting new insights into nucleation mechanisms, considering pure phases of anisotropic molecules, and the transition from nematic to smectic ordering upon temperature quenching (which is a weakly first-order phase transition as corroborated by the present study). Research in this area is highly topical and important, as the authors convincingly summarize in the introduction. The work is of high quality, and overall suitable for publication in Nature Communications. However, before I can recommend acceptance, some points have to be addressed and resolved. These might not necessarily require additional calculations and simulations but since some of the concerns are substantial, I recommend acceptance after major revisions as noted.

Major points

-It appears that the authors have not explored the dependence of the nucleation pathway on the degree of supercooling, which, in my opinion, would provide crucial further insights into the mechanism by elucidating the scaling of the barriers and local minima with the driving force for phase separation. Do they scale similarly, or do certain characteristics even exist only at specific driving force? What is the dependence of the height of barriers (and depth of local minima) on the level of supercooling?

The authors should justify, thoroughly, in case they object with this point.

- In the free energy profiles, metastable clusters at $N=7$ and $N=150$ occur, associated with barriers on the order of kT and $2kT$ but the as-discussed local minima and “wrinkles” are MUCH smaller than that. Given that kT (thermal energy) is comparable to typical errors from quite high levels of quantum mechanical theory and of the order of so-called chemical accuracy, I wonder whether the found features are really significant. What is the error of the calculations, do the authors just discuss noise here? The exploration of the scaling of these features with the level of supercooling (see above) might allow increasing the readers’ confidence that indeed, significant characteristics are investigated and discussed.

- Lines 136 ff.; “MC1 and MC2 with distinct smectic layers can thus be regarded as pre-nucleation clusters (PNCs) in the sense that they are dynamic entities with structural motifs resembling the bulk smectic phase, possessing the characteristics of PNCs summarized in Ref. 16.”

It appears that the MC1 and MC2 species does not fully qualify as PNCs within the definitions provided in Ref. 16. While they seem to possess some structural characteristics of the final phase that occur in PNCs, PNCs are defined as thermodynamically stable solute clusters (homo phase), please also see the minor point on the distinction between PNCs and two-step nucleation below. However, MC1 and MC2 are metastable (and phase-separated, thus, not strictly “pre-nucleation?”).

The authors should clarify in which characteristics the found species agree with the definition of PNCs and in which not. While the formation of PNCs is not associated with major barriers, by definition in ref 16, the barriers are quite small for MC1 and MC2 (see above), as expected for a weakly first-order transition, and they might thus well qualify as “metastable PNCs”. However, again, the elucidation of the scaling of the (small barriers) with the degree of supercooling would allow to comment on the commonalities and differences with PNCs in more detail, rather than in a handwaving manner as in the current manuscript.

Please clarify.

-Lines 166 ff; the authors discuss the free energy of the clusters in relation to their existence probability. As the free energy is positive, the existence probability is very low. How can those species then be detected experimentally, e.g., by scattering as discussed by the authors? What would be the total scattering intensity for a realistic pre-transitional situation? Can the described mechanism thus really explain the pre-transitional clusters observed in experiments?

Please clarify.

Minor points

- Line 37 ff.; "A nucleus starts to grow when its size that obeys the Boltzmann distribution exceeds a critical value determined by the balance of the bulk and the surface free energies."

The correct spelling is "Boltzmann".

I agree that the population of the clusters of different sizes obeys Boltzmann statistics, according to classical nucleation theory, i.e., according to their respective free energies. The latter does not depend on cluster size in a linear fashion. So, strictly speaking, it is not the size but the cluster populations that obey Boltzmann statistics. Please comment and clarify in the manuscript as required.

-In summarizing the notion of "two step nucleation", sometimes used to explain discrepancies between classical nucleation theory and experiment, the authors also cite papers dealing with the so called "pre-nucleation cluster pathway". However, the two non-classical mechanisms are distinct from a physical chemical viewpoint. In a nutshell, PNCs are solute (homo phase) precursors that are thermodynamically stable within the boundary of the mother solution. Their formation is not considered to be associated with any considerable barriers, so they always form, independent of supersaturation. Once the mother solution becomes supersaturated (metastable), PNCs can transform into phase separated entities due to distinctly reduced dynamics. These post-nucleation species thus formed directly from PNCs are then metastable with respect to crystals and can subsequently transform (ripening). By contrast, in "two-step nucleation" the observed intermediates form through metastable fluctuations and only upon exceeding a certain supersaturation threshold, as their formation is associated with a barrier. A more detailed distinction is discussed e.g. in doi: 10.1016/j.nantod.2011.10.005

The authors should briefly clarify that the mechanism are different, and thereby potentially take the role of supersaturation (supercooling), and the fact that the transition discussed here is weakly first-order, into account.

-Line 90 "...and its saturation at $t \geq 1.5\tau$ indicates that the percolation of smectic phase progresses at..."

It appears that the number of smectic molecules does never saturate even as tau approaches 5 (the maximum on the abscissa).

Please clarify.

Reviewer #2 (Remarks to the Author):

The authors study the nucleation and growth in the nematic to smectic transition in the Gay Berne liquid. They use a machine learning algorithm from previous work to identify smectic nuclei and follow their growth in simulations of large systems. They find three steps, an initial unproductive nucleation, followed by a two step nucleation that leads to the phase transition. The system size is sufficiently large (1 Million particles) which is important because the nuclei are composed hundreds of particles.

I thought the paper was interesting, but did not think it was of general interest. It is not clear if these observations are general and applicable to other systems. I therefore thought the paper was interesting by more appropriate to a specialized journal on computer simulations or liquid crystals.

Reviewer #3 (Remarks to the Author):

The manuscript by Takahashi et al. reports on the multistep nucleation of a smectic phase from a nematic fluid using large scale MD simulations of anisotropic molecules. The work is interesting and appears to have been thoroughly carried out and it suggests the existence of a complicated, non-classical, pathway for the nucleation of the smectic phase and that stable pre-nucleation clusters exist along that path.

In the last decade there has been a fairly hot debate about two/multi-step nucleation and about

the existence of stable pre-nucleation clusters before the onset of nucleation of mineral phases, particularly CaCO_3 . Although this debate was covered fairly well in the introduction and discussion, I think the use of the term pre-nucleation clusters (PNCs) in the context of a nematic/smectic phase transition is somewhat misleading and I reckon the present manuscript won't be very relevant to the geochemistry community.

In fact, the notion of pre-nucleation clusters that was introduced by Gebauer et al. in 2008 for CaCO_3 has a somewhat different meaning to the one implied in this manuscript. CaCO_3 PNCs have been suggested to be thermodynamically stable entities with respect to the isolated ions in solution i.e. they have a lower free energy than the isolated ions. On the other hand, in this paper PNC refers to structures that have a higher free energy than the nematic phase they originated from; and even if though they are meta-stable states on the free energy landscape, one can envisage that they are formed after a nucleation event. Therefore, I think it would be more appropriate to discuss the results reported here in terms of multi-step nucleation, i.e. a sequence of (possibly) classical nucleation events, and drop the term pre-nucleation clusters.

Moreover, because the nematic/smectic phase transition is different from the nucleation of an ionic crystal from solution, I would recommend the authors to refocus the introduction and discussion of their manuscript to cover the literature on the nucleation from the melt; either for model systems (such as Lennard-Jones fluids), molten salts (e.g. NaCl) or molecular solids (e.g. water), where there is also ample debate about whether classical nucleation theory works or not.

I found the discussion about the supercritical nuclei very convoluted and hard to follow, so much so that it wasn't clear to me what it was actually showing. The paper would greatly benefit from expanding this section or making clearer why that analysis is different/better from the results shown in figure 1 where the nucleation barrier is shown.

The authors should make clear in their manuscript what is the zero of the free energy reported in figure 1e and the SI. From the inset it seems that clusters composed of one molecule already have a free energy of $9k_B T$. Hence, the rate limiting step for the nucleation of the smectic phase would be the formation of smectic clusters composed of one molecule, which I imagine it means that the central molecule in the cluster is in an environment that resemble the smectic phase, while its nearest neighbours aren't?

Have the authors repeated their simulations at different levels of undercooling to verify whether the nucleation path depends on the temperature? Will they see a spinodal transition?

It seems that the molecules in the movie included as SI are not changing orientation, but just appearing/disappearing or changing colours while I expected the movie to be showing an MD trajectory. Is this just an optical illusion? How was the movie generated?

Reply to comments of Reviewer 1

Reviewer 1 Comment 1:

The manuscript of Takahashi et al is very well written, dealing with the multistep nucleation of anisotropic molecules, which was explored computationally. Namely, the authors used a combination of molecular dynamics simulation, machine learning, and molecular cluster analysis to obtain very interesting new insights into nucleation mechanisms, considering pure phases of anisotropic molecules, and the transition from nematic to smectic ordering upon temperature quenching (which is a weakly first-order phase transition as corroborated by the present study).

Research in this area is highly topical and important, as the authors convincingly summarize in the introduction. The work is of high quality, and overall suitable for publication in Nature Communications. However, before I can recommend acceptance, some points have to be addressed and resolved. These might not necessarily require additional calculations and simulations but since some of the concerns are substantial, I recommend acceptance after major revisions as noted.

Major points

-It appears that the authors have not explored the dependence of the nucleation pathway on the degree of supercooling, which, in my opinion, would provide crucial further insights into the mechanism by elucidating the scaling of the barriers and local minima with the driving force for phase separation. Do they scale similarly, or do certain characteristics even exist only at specific driving force? What is the dependence of the height of barriers (and depth of local minima) on the level of supercooling?

The authors should justify, thoroughly, in case they object with this point.

Reply to Comment 1:

Let us first thank Reviewer 1 for high evaluation of the quality of our work. We also appreciate his/her valuable comments, suggestions and criticisms that helped us to improve the quality of our work and presentation as we describe below.

Investigation of how the nucleation pathway depends on the degree of supercooling (denoted by ΔT below) is indeed important and was missing in our initial submission. We greatly appreciate the suggestion of the reviewer. We carried out further systematic calculations to elucidate the effect of the variation of ΔT . As summarized below, we have made additional findings that clearly indicate that the free energy pathway is markedly distinct from that expected by the classical nucleation theory. The findings are presented in Lines 144-161 of page 7 and in Figs. 2, S5, and S6. We believe that the additional statements on the effect of the variation of the supercooling have significantly improved the quality of our work, and again we thank the reviewer for a valuable suggestion.

(Summary of our findings on the effect of the variation of ΔT)

We confirmed that the important features of the free energy landscape are insensitive to ΔT (see Fig. S5). We also clarified the ΔT dependence of the sizes of characteristic clusters, MC1, MC2, and CN. The size of CN is roughly proportional to ΔT^{-1} , clearly different from the scaling law of classical nucleation theory $\propto \Delta T^{-3}$ (see Fig. 2 (a)). The sizes of MC1 and MC2 sizes are almost constant (respectively 7 and 168 ± 20 , see Fig. S6 (a)). We also clarified the ΔT dependence of the height of energy barriers, and found that the height of the barrier from nematic to CN is proportional to $\ln(\Delta T^{-1.4})$, demonstrating again the difference from the behavior expected from the classical nucleation $\propto \Delta T^{-2}$ (see Fig. 2 (b)). The barrier from MC1 to MC2 decreases weakly with the increase of ΔT and falls within the range [$1.60k_B T$, $2.22k_B T$] (see Fig. S6 (b)). The barriers from nematic to MC1 and MC2 to CN are almost constant ($(9.68 \pm 0.04)k_B T$ and $(0.77 \pm 0.08)k_B T$, respectively). The depth of the local minimum of MC1 and MC2 is only weakly dependent of ΔT and increases with the increase of ΔT (see Fig. S6 (c)).

Reviewer 1 Comment 2:

- In the free energy profiles, metastable clusters at N=7 and N=150 occur, associated with barriers on the order of kT and 2kT but the as-discussed local minima and “wrinkles” are MUCH smaller than that. Given that kT (thermal energy) is comparable to typical errors from quite high levels of quantum mechanical theory and of the order of so-called chemical accuracy, I wonder whether the found features are really significant. What is the error of the calculations, do the authors just discuss noise here? The exploration of the scaling of these features with the level of supercooling (see above) might allow increasing the readers’ confidence that indeed, significant characteristics are investigated and discussed.

Reply to Comment 2:

We appreciate the reviewer’s comment on the accuracy of the calculation. The high accuracy of the method for computing free energy landscapes has been demonstrated in Refs. 48 and 55, on which our analysis of the free energy landscape is based. As mentioned in Reply to Comment 1, our additional calculations successfully identified the free energy landscape exhibiting MC1 and MC2 in a wide range of ΔT , which we believe reflects the reliability of our calculations as well as the robustness of the formation of MC1 and MC2.

Reviewer 1 Comment 3:

- Lines 136 ff.; “MC1 and MC2 with distinct smectic layers can thus be regarded as pre-nucleation clusters (PNCs) in the sense that they are dynamic entities with structural motifs resembling the bulk smectic phase, possessing the characteristics of PNCs summarized in Ref. 16.”

It appears that the MC1 and MC2 species does not fully qualify as PNCs within the definitions provided in Ref. 16. While they seem to possess some structural characteristics of the final phase that occur in PNCs, PNCs are defined as thermodynamically stable solute clusters (homo phase), please also see the minor point on the distinction between PNCs and two-step nucleation below. However, MC1 and MC2 are metastable (and phase-separated, thus, not strictly “pre-nucleation?”).

The authors should clarify in which characteristics the found species agree with the definition of PNCs and in which not. While the formation of PNCs is not associated with major barriers, by definition in ref 16, the barriers are quite small for MC1 and MC2 (see above), as expected for a weakly first-order transition, and they might thus well qualify as “metastable PNCs”. However, again, the elucidation of the scaling of the (small barriers) with the degree of supercooling would allow to comment on the commonalities and differences with PNCs in more detail, rather than in a handwaving manner as in the current manuscript.

Please clarify.

Reply to Comment 3:

We appreciate the reviewer’s valuable comment on pre-nucleation clusters. After carefully reading the comments of Reviewer 1 and also those of Reviewer 3, we now think it is not a good idea to put too much emphasis on the similarity of our metastable clusters (MC1 and MC2) to pre-nucleation clusters. Therefore, we rewrote the abstract, and the sections “Real-space density profiles of metastable clusters and critical nuclei” (Page 8, Line 176) and “Discussion” (Pages 11-12, Lines 255-279) to eliminate our previous claim that our MC1 and MC2 resemble PNCs.

As we commented in “Reply to Comment 1”, we carried out a systematic analysis on the effect of the variation of supercooling ΔT . From the low energy barrier from MC1 ($\sim 2.0k_B T$), and the size insensitivity to ΔT , MC2 might be regarded as “metastable PNC” as the reviewer suggests, and we commented on this feature of MC2 at the end of “Free energy landscape” (Page 7, Lines 158-160).

Reviewer 1 Comment 4:

-Lines 166 ff; the authors discuss the free energy of the clusters in relation to their existence probability. As the free energy is positive, the existence probability is very low. How can those species then be detected experimentally, e.g., by scattering as discussed by the authors? What would be the total scattering intensity for a realistic pre-transitional situation? Can the described mechanism thus really explain the pre-transitional clusters observed in experiments?

Please clarify.

Reply to Comment 4:

We appreciate a valuable suggestion to clarify the relevance of our findings to experiments. We first note that the detection of pre-transitional fluctuations (or cybotactic clusters) is still an experimental

challenge. In X-ray scattering experiments (49,50, R1, R2), the average density of pre-transitional fluctuations is determined from the peak value of scattering intensity, which gives only a rough estimation and cannot discern different types of smectic clusters (Still, our Figure 1b clearly shows strong correlation between the scattering intensity and the number of smectic molecules). Cryo-TEM observations have acquired real-space molecular arrangements of smectic clusters, but information on the dynamics is lost.

Moreover, most of the recent cryo-TEM studies concern bent-core LCs because cybotactic clusters (or cybotactic nematic “phase”) are much more easily observed than in conventional LCs (50,R2). The smectic phase of the bent-core LCs used in the experiments is smectic-C with the orientation of molecules tilted from the smectic layer normal, while our numerical system exhibited smectic A (in which the molecular orientation is normal to the smectic layers) (Figure S2).

Although we believe that our calculation based on a simple soft-core Gay-Berne model captures the essential features of smectic nucleation as described in the manuscript, specific properties arising from the bent molecular shape need further investigation for direct comparison with experimental results. The effect of molecular shape is indeed an important and interesting subject, but we would like to leave it for future studies because a substantial amount of numerical resource and time will be necessary for additional calculations.

[R1] O. Francescangeli and E. T. Samulski, “Insights into the cybotactic nematic phase of bent-core molecules”, *Soft Matter*, 6, 2413-2420 (2010).

[R2] O. Francescangeli, F. Vita and E. T. Samulski, “The cybotactic nematic phase of bent-core mesogens: state of the art and future developments”, *Soft Matter*, 10, 7685 (2014).

Reviewer 1 Comment 5:

Minor points

- Line 37 ff.; “A nucleus starts to grow when its size that obeys the Boltzmann distribution exceeds a critical value determined by the balance of the bulk and the surface free energies.”

The correct spelling is “Boltzmann”.

I agree that the population of the clusters of different sizes obeys Boltzmann statistics, according to classical nucleation theory, i.e., according to their respective free energies. The latter does not depend on cluster size in a linear fashion. So, strictly speaking, it is not the size but the cluster populations that obey Boltzmann statistics. Please comment and clarify in the manuscript as required.

Reply to Comment 5:

We thank the reviewer for carefully reading our manuscript and pointing out our mistake. We agree with the reviewer’s comment and have rewritten the corresponding part (Page 2-3, Lines 37-40).

Reviewer 1 Comment 6:

-In summarizing the notion of “two step nucleation”, sometimes used to explain discrepancies between classical nucleation theory and experiment, the authors also cite papers dealing with the so called “pre-nucleation cluster pathway”. However, the two non-classical mechanisms are distinct from a physical chemical viewpoint. In a nutshell, PNCs are solute (homo phase) precursors that are thermodynamically stable within the boundary of the mother solution. Their formation is not considered to be associated with any considerable barriers, so they always form, independent of supersaturation. Once the mother solution becomes supersaturated (metastable), PNCs can transform into phase separated entities due to distinctly reduced dynamics. These post-nucleation species thus formed directly from PNCs are then metastable with respect to crystals and can subsequently transform (ripening). By contrast, in “two-step nucleation” the observed intermediates form through metastable

fluctuations and only upon exceeding a certain supersaturation threshold, as their formation is associated with a barrier. A more detailed distinction is discussed e.g. in doi:

10.1016/j.nantod.2011.10.005

The authors should briefly clarify that the mechanism are different, and thereby potentially take the role of supersaturation (supercooling), and the fact that the transition discussed here is weakly first-order, into account.

Reply to Comment 6:

We again thank the reviewer for valuable comments on PNCs, and have rewritten the Introduction (Page 3, Lines 47-55) to clarify the properties of PNCs and their difference from two-step nucleation. Reference 21 (doi: 10.1016/j.nantod.2011.10.005) has been added and cited there.

Reviewer 1 Comment 7:

-Line 90 “...and its saturation at $t \geq 1.5\tau$ indicates that the percolation of smectic phase progresses at...”

It appears that the number of smectic molecules does never saturate even as tau approaches 5 (the maximum on the abscissa).

Please clarify.

Reply to Comment 7:

We appreciate the comment, and apologize for the ambiguity of “saturation” there. We meant that the number of smectic molecules belonging to the largest cluster becomes almost equal to the total number of smectic molecules in the system, indicating the onset of percolation process during/after nuclear growth. To make it clear, we have rephrased the sentence (Page 5, Lines 100-101). Note that

the saturation of the total number of smectic molecules in the system is the slowest process in phase transition, and is beyond the scope of this work.

Reply to comments of Reviewer 2

Reviewer 2 Comment 1:

The authors study the nucleation and growth in the nematic to smectic transition in the Gay Berne liquid. They use a machine learning algorithm from previous work to identify smectic nuclei and follow their growth in simulations of large systems. They find three steps, an initial unproductive nucleation, followed by a two step nucleation that leads to the phase transition. The system size is sufficiently large (1 Million particles) which is important because the nuclei are composed hundreds of particles.

I thought the paper was interesting, but did not think it was of general interest. It is not clear if these observations are general and applicable to other systems. I therefore thought the paper was interesting by more appropriate to a specialized journal on computer simulations or liquid crystals.

Reply to Comment 1:

We regret that Reviewer 2 regarded our work as “more appropriate to a specialized journal.” Considering the long history of research on nucleation still under hot debate, we believe that our demonstration of non-trivial multi-step nucleation in a simple model system of anisotropic molecules will not only appeal to audience in the specific fields of computer simulations or liquid crystals, but also draw broad interest from diverse research fields including condensed matter physics, chemical physics, materials science and biological science. We also emphasize that the simplicity of our model molecules rather enables us to claim the universality of our findings (Indeed, we have carried out calculations using a different model of anisotropic molecules to check that our findings are not model-specific: See the second paragraph of “Free energy landscape”). Moreover, additional calculations on the effect of the variation of supercooling, suggested by Reviewers 1 and 3, have significantly improved the quality of our work. We therefore believe that our work merits publication in *Nature Communications*, rather than in a specialized journal.

Reply to comments of Reviewer 3

Reviewer 3 Comment 1:

The manuscript by Takahashi et al. reports on the multistep nucleation of a smectic phase from a nematic fluid using large scale MD simulations of anisotropic molecules. The work is interesting and appears to have been thoroughly carried out and it suggests the existence of a complicated, non-classical, pathway for the nucleation of the smectic phase and that stable pre-nucleation clusters exist along that path.

In the last decade there has been a fairly hot debate about two/multi-step nucleation and about the existence of stable pre-nucleation clusters before the onset of nucleation of mineral phases, particularly CaCO₃. Although this debate was covered fairly well in the introduction and discussion, I think the use of the term pre-nucleation clusters (PNCs) in the context of a nematic/smectic phase transition is somewhat misleading and I reckon the present manuscript won't be very relevant to the geochemistry community.

In fact, the notion of pre-nucleation clusters that was introduced by Gebauer et al. in 2008 for CaCO₃ has a somewhat different meaning to the one implied in this manuscript. CaCO₃ PNCs have been suggested to be thermodynamically stable entities with respect to the isolated ions in solution i.e. they have a lower free energy than the isolated ions. On the other hand, in this paper PNC refers to structures that have a higher free energy than the nematic phase they originated from; and even if though they are meta-stable states on the free energy landscape, one can envisage that they are formed after a nucleation event. Therefore, I think it would be more appropriate to discuss the results reported here in terms of multi-step nucleation, i.e. a sequence of (possibly) classical nucleation events, and drop the term pre-nucleation clusters.

Reply to Comment 1:

We first appreciate Reviewer 3's high evaluation of our work saying that it "is interesting and appears to have been thoroughly carried out." We also thank him/her for valuable comments, suggestions and criticisms that helped us to improve the quality of our work and presentation as we describe below.

We appreciate the reviewer's valuable comment on pre-nucleation clusters. As we commented in Reply to Comment 3 of Reviewer 1, now we now think it is not a good idea to put too much emphasis on the similarity of our metastable clusters (MC1 and MC2) to pre-nucleation clusters. Therefore, we rewrote the abstract, and the sections "Real-space density profiles of metastable clusters and critical nuclei" (Page 8, Line 176) and "Discussion" (Pages 11-12, Lines 255-279) to eliminate our previous claim that our MC1 and MC2 resemble PNCs.

As we commented in Reply to Comments 1 and 3 of Reviewer 1, we carried out a systematic analysis on the effect of the variation of supercooling ΔT . From the low energy barrier from MC1 ($\sim 2.0k_B T$), and the size insensitivity to ΔT , MC2 might be regarded as “metastable PNC” as Reviewer 1 suggests, and we commented on this feature of MC2 at the end of “Free energy landscape” (Page 7, Lines 158-160). Still, as suggested by the reviewer, it is more appropriate to interpret our results in the context of multistep nucleation. Again we thank the reviewer for motivating us to reconsider the our results in the context of multistep nucleation.

Reviewer 3 Comment 2:

Moreover, because the nematic/smectic phase transition is different from the nucleation of an ionic crystal from solution, I would recommend the authors to refocus the introduction and discussion of their manuscript to cover the literature on the nucleation from the melt; either for model systems (such as Lennard-Jones fluids), molten salts (e.g. NaCl) or molecular solids (e.g. water), where there is also ample debate about whether classical nucleation theory works or not.

Reply to Comment 2:

We thank the reviewer for a valuable suggestion, according to which we have written the Introduction to clarify classical, two-step, and more complex nucleation scenarios (Page 3 Lines 47-55). In the previous version we had already quoted some references on model systems that can be understood within the context of the classical nucleation theory (see Refs.14,15,35,36,38,39). As we will mention in Reply to Comment 5 below (or we have had in Reply to Comment 1 of Reviewer 1), additional calculations focusing on the effect of the variation of the degree of undercooling reconfirmed the non-classical nature of smectic nucleation, as discussed in “Free energy landscape” (Page 7, Lines 144-161).

Reviewer 3 Comment 3:

I found the discussion about the supercritical nuclei very convoluted and hard to follow, so much so that it wasn't clear to me what it was actually showing. The paper would greatly benefit from expanding this section or making clearer why that analysis is different/better from the results shown in figure 1 where the nucleation barrier is shown.

Reply to Comment 3:

We regret that our previous presentation was unclear and confused the reviewer. We have written the first two paragraphs of the section “Dynamics of metastable clusters, critical nuclei and supercritical nuclei in the transition” to make clearer the motivation of our analysis there and the definition of the symbols for the clusters (Pages 8-9, Lines 183-205). We would like to emphasize that the analysis of

the actual dynamics of smectic clusters in addition to the elucidation of the free energy landscape deepens the understanding of smectic nucleation and further clarifies its non-classical nature. We thank the reviewer for his/her suggestion to improve the presentation there.

Reviewer 3 Comment 4:

The authors should make clear in their manuscript what is the zero of the free energy reported in figure 1e and the SI. From the inset it seems that clusters composed of one molecule already have a free energy of $9k_B T$. Hence, the rate limiting step for the nucleation of the smectic phase would be the formation of smectic clusters composed of one molecule, which I imagine it means that the central molecule in the cluster is in an environment that resembles the smectic phase, while its nearest neighbours aren't?

Reply to Comment 4:

The reference (zero) of the free energy was taken to be that of the nematic phase (Page 7, Line 139), but we should have made it clear much earlier when we first introduced Figure 1(c) and (e). We have clarified it in Lines 120-121 of page 6, and we thank the reviewer for drawing our attention to this unclear presentation.

For the “smectic clusters composed of one molecule”, the understanding of the reviewer is correct. The local order parameter developed by us can classify whether one given molecule belongs to a nematic phase or a smectic phase, and as the reviewer mentions, “smectic clusters composed of one molecule” is one (central) molecule that satisfies the criteria of smectic order, while neighbouring ones do not. We also note that local order parameters of a similar kind have already been successfully employed for the classification of a wide variety of molecular structures such as ice polymorphs and polymer crystals[38,51,54,68,69,R3-5].

[R3] W. Lechner and C. Dellago, “Accurate determination of crystal structures based on averaged local bond order parameters,” *The Journal of Chemical Physics*, 129, 114707 (2008).

[R4] A. Reinhardt, J. P. K. Doye, E. G. Noya, and C. Vega, “Local order parameters for use in driving homogeneous ice nucleation with all-atom models of water,” *The Journal of Chemical Physics* 137, 194504 (2012).

[R5] X. Tang, J. Yang, T. Xu, F. Tian, C. Xie, and L. Li, “Local structure order assisted two-step crystal nucleation in polyethylene,” *Physical Review Materials*, 1, 073401 (2017).

Reviewer 3 Comment 5:

Have the authors repeated their simulations at different levels of undercooling to verify whether the nucleation path depends on the temperature? Will they see a spinodal transition?

Reply to Comment 5:

We appreciate the reviewer's valuable suggestion for the investigation of the effect of the variation of the degree of undercooling (hereafter denoted by ΔT). The same suggestion was given by Reviewer 1, and as we commented in Reply to Comment 1 of Reviewer 1, we carried out systematic calculations to elucidate the effect of the variation of ΔT . As summarized below, we have made additional findings that clearly indicate that the free energy pathway is markedly distinct from that expected by the classical nucleation theory. The findings are presented in Lines 144-161 of page 7 and in Figs. 2, S5, and S6. We believe that the additional statements on the effect of the variation of the undercooling have significantly improved the quality of our work, and again we thank the reviewer for a valuable suggestion. We also comment that our results clearly indicate the weakly-first-order nature of the nematic-smectic phase transition consistent with previous studies, and that spinodal decomposition was not observed.

(Summary of our findings on the effect of the variation of ΔT)

We confirmed that the important features of the free energy landscape are insensitive to ΔT (see Fig. S5). We also clarified the ΔT dependence of the sizes of characteristic clusters, MC1, MC2, and CN. The size of CN is roughly proportional to ΔT^{-1} , clearly different from the scaling law of classical nucleation theory $\propto \Delta T^{-3}$ (see Fig. 2 (a)). The sizes of MC1 and MC2 sizes are almost constant (respectively 7 and 168 ± 20 , see Fig. S6 (a)). We also clarified the ΔT dependence of the height of energy barriers, and found that the height of the barrier from nematic to CN is proportional to $\ln(\Delta T^{-1.4})$, demonstrating again the difference from the behavior expected from the classical nucleation $\propto \Delta T^{-2}$ (see Fig. 2 (b)). The barrier from MC1 to MC2 decreases weakly with the increase of ΔT and falls within the range [$1.60k_B T$, $2.22k_B T$] (see Fig. S6 (b)). The barriers from nematic to MC1 and MC2 to CN are almost constant ($(9.68 \pm 0.04)k_B T$ and $(0.77 \pm 0.08)k_B T$, respectively). The depth of the local minimum of MC1 and MC2 is only weakly dependent of ΔT and increases with the increase of ΔT (see Fig. S6 (c)).

Reviewer 3 Comment 6:

It seems that the molecules in the movie included as SI are not changing orientation, but just appearing/disappearing or changing colours while I expected the movie to be showing an MD trajectory. Is this just an optical illusion? How was the movie generated?

Reply to Comment 6:

The caption of Movie S1 may have not been clear and we apologize for that. Only molecules belonging to MC1, MC2, CN, tMC1, tMC2, and tCN are displayed. Molecules that belong to none of them are not depicted. Colour change of molecules means that the type of clusters the molecules

belong to has been changed, and the disappearance means that the molecules no longer belong to any of the clusters. The caption has been written for clarification. We also note that during the nematic-smectic transition, the direction of liquid crystal molecules does not change significantly.

REVIEWERS' COMMENTS

Reviewer #1 (Remarks to the Author):

The authors have resolved my previous concerns and have carefully revised the manuscript. I strongly recommend publication of this work in Nat Commun.

Reply to comments of Reviewer 1

Reviewer 1 Comment 1:

The authors have resolved my previous concerns and have carefully revised the manuscript. I strongly recommend publication of this work in Nat Commun.

Reply to Comment 1:

We thank the reviewer for the high appreciation of the revised manuscript, and the valuable comments and suggestions throughout the review process which greatly helped improve the quality of our manuscript.